

# Parallel power posterior analyses for fast computation of marginal likelihoods in phylogenetics

Sebastian Höhna[1,2], Michael J. Landis[3] and John P. Huelsenbeck[4]

[1] GeoBio-Center, Ludwig-Maximilians-Universität München, Munich, Germany
[2] Department of Earth and Environmental Sciences, Paleontology & Geobiology, Ludwig-Maximilians-Universität München, Munich, Germany
[3] Department of Biology, Washington University in St. Louis, St. Louis, United States of America
[4] Department of Integrative Biology, University of California,, Berkeley, United States of America

## ABSTRACT

In Bayesian phylogenetic inference, marginal likelihoods can be estimated using several different methods, including the path-sampling or stepping-stone-sampling algorithms. Both algorithms are computationally demanding because they require a series of power posterior Markov chain Monte Carlo (MCMC) simulations. Here we introduce a general parallelization strategy that distributes the power posterior MCMC simulations and the likelihood computations over available CPUs. Our parallelization strategy can easily be applied to any statistical model despite our primary focus on molecular substitution models in this study. Using two phylogenetic example datasets, we demonstrate that the runtime of the marginal likelihood estimation can be reduced significantly even if only two CPUs are available (an average performance increase of 1.96x). The performance increase is nearly linear with the number of available CPUs. We record a performance increase of 13.3x for cluster nodes with 16 CPUs, representing a substantial reduction to the runtime of marginal likelihood estimations. Hence, our parallelization strategy enables the estimation of marginal likelihoods to complete in a feasible amount of time which previously needed days, weeks or even months. The methods described here are implemented in our open-source software RevBayes which is available from http://www.RevBayes.com.

# INTRODUCTION

Model selection in Bayesian phylogenetic inference is performed by computing Bayes factors, which are ratios of the marginal likelihoods for two alternative models (*Kass & Raftery, 1995*; *Sullivan & Joyce, 2005*). The Bayes factor indicates support for a model when the ratio of the marginal likelihoods is greater than one. This procedure is very similar to likelihood ratio tests with the difference being that one averages the likelihood over all possible parameter values weighted by the prior probability rather than maximizing the likelihood with respect to the parameters (*Posada & Crandall, 2001*; *Holder & Lewis, 2003*). More specifically, the marginal likelihood of a model, $f(D|M)$, is calculated as the product of the likelihood, $f(D|\theta, M)$, and the prior, $f(\theta|M)$, integrated (or marginalized) over all

Corresponding author
Sebastian Höhna, hoehna@lmu.de

possible parameter combinations,

$$f(D|M) = \int f(D|\theta, M) f(\theta|M) d\theta. \tag{1}$$

In the context of Bayesian phylogenetic inference, this quantity is computed by marginalizing over the entire parameter space, namely over all possible tree topologies, branch lengths, substitution model parameters and other model parameters (*Huelsenbeck et al., 2001*; *Suchard, Weiss & Sinsheimer, 2001*).

The computation of the marginal likelihood is intrinsically difficult because the dimension-rich integral is impossible to compute analytically (*Oaks et al., 2019*). Monte Carlo sampling methods have been proposed to circumvent the analytical computation of the marginal likelihood (*Gelman & Meng, 1998*; *Neal, 2000*). *Lartillot & Philippe (2006)* introduced a technique called thermodynamic integration, (also called path-sampling; *Baele et al., 2012a*), to approximate the marginal likelihood. A similar method, stepping-stone-sampling (*Xie et al., 2011*; *Fan et al., 2011*), has more recently been proposed (see also *Baele et al., 2012a*; *Baele & Lemey, 2013*; *Friel, Hurn & Wyse, 2014*; *Oaks et al., 2019*; *Fourment et al., 2020* for a summary and comparison of these methods). The fundamental idea of path-sampling and stepping-stone-sampling is to use a set of $K$ importance distributions, or power posterior distributions, from which likelihood samples are taken (*Gelman & Meng, 1998*; *Neal, 2000*; *Lartillot & Philippe, 2006*; *Friel & Pettitt, 2008*). The sampling procedure for each importance distribution is performed by a Markov chain Monte Carlo (MCMC) algorithm. That is, instead of running a single MCMC simulation, as is commonly done to estimate posterior probabilities (*Huelsenbeck et al., 2001*; *Huelsenbeck et al., 2002*), $K$ (usually between $K = 30$ and $K = 200$) MCMC simulations are needed to estimate the marginal likelihood of a model of interest (*Fan et al., 2011*). Estimating marginal likelihoods for phylogenetic models using power posterior distributions is implemented, amongst others, in MrBayes (*Ronquist et al., 2012*), PhyloBayes (*Lartillot, Lepage & Blanquart, 2009*), BEAST 1 & 2 (*Suchard et al., 2018*; *Bouckaert et al., 2019*) and Phycas (*Lewis, Holder & Swofford, 2015*). Obviously, this strategy can be very time consuming considering that a single MCMC simulation may take from hours to several weeks of computer time. The high computational time poses a major challenge for Bayes factor computations for many important problems, for example, comparing molecular substitution models (*Posada & Crandall, 2001*), selecting between complex diversification rate models (*FitzJohn, 2012*), and evaluating competing continuous trait processes (*e.g.*, *Uyeda & Harmon, 2014*).

In the present article we demonstrate how power posterior simulations can be performed on parallel computer architectures and report the achieved computational gain. The idea of parallel power posterior simulations is very similar to parallel Metropolis coupled MCMC algorithm (*Altekar et al., 2004*), with the important difference that power posterior simulations can be parallelized even more easily because no communication between processes is necessary. Additionally we show how our parallelization scheme can combined with existing parallelization techniques for distributed likelihood computation (*e.g.*, *Aberer, Kobert & Stamatakis, 2014*) to maximize usage of available CPUs. Here we focus on two common approaches of computing marginal likelihoods: path-sampling and

stepping-stone-sampling. Nevertheless, our parallelization strategy is also applicable to other approaches such as the generalized stepping-stone-sampler (GSS) (*Fan et al., 2011*; *Holder et al., 2014*; *Baele, Lemey & Suchard, 2016*).

## METHODS

The algorithm underlying path-sampling and stepping-stone-sampling can be separated into two steps: (1) likelihood samples are obtained from a set of $K$ power posterior simulations; and (2) the marginal likelihood is approximated either by numerical integration of the likelihood samples over the powers (path-sampling) or by the likelihood ratio between powers (stepping-stone-sampling). The first step is the same for both methods and is the computationally expensive part. Thus, once samples from the power posterior distributions are obtained, it is possible to rapidly compute both the path-sampling and stepping-stone-sampling marginal likelihood estimates.

### Power posterior sampling

Both stepping-stone-sampling and path-sampling techniques construct and sample from a series of importance distributions. *Lartillot & Philippe (2006)* define the importance distributions as power posterior distributions, which are obtained by modifying the posterior probability density as

$$f_{\beta_i}(\theta) = f(Y|\theta, M)^{\beta_i} f(\theta|M). \tag{2}$$

Here, $\beta$ represent a vector of powers between 0 and 1. Then, for every value of $\beta_i$ a draw from the power posterior distribution is needed and its likelihood score, $l_i$, is recorded (*Lartillot & Philippe, 2006*; *Friel & Pettitt, 2008*). In principle, one such likelihood sample per power posterior distribution is sufficient, although multiple samples improve the accuracy of the estimated marginal likelihood considerably (*Baele et al., 2012b*; *Oaks et al., 2019*; *Fourment et al., 2020*). We will use the notation $l_{ij}$ to represent the $j$th likelihood sample from the $i$th power posterior distribution.

We illustrate the mean log-likelihood over different values of $\beta$ in Fig. 1. Commonly, the values of the powers $\beta$ are set to the $i$th quantile of a beta$(0.3, 1.0)$ distribution (*Xie et al., 2011*; *Baele et al., 2012a*). The rationale is that more narrowly spaced intervals are needed for the range of $\beta$ where the expected likelihood changes most rapidly, *i.e.*, for $\beta$ values close to 0 (Fig. 1).

Draws from the power posterior distribution are obtained by running a modified Markov chain Monte Carlo (MCMC *Metropolis et al., 1953*; *Hastings, 1970*) algorithm:
1. Let $\theta_j$ denote the current parameter values at the $j$th iteration, initialized at random at the start of the MCMC algorithm.
2. Propose a new values $\theta'$ drawn from a proposal kernel with density $q(\theta'|\theta_j)$.
3. The proposed state is accepted with probability
$$\alpha = \min\left(1, \frac{f(D|\theta')^{\beta_i}}{f(D|\theta_j)^{\beta_i}} \times \frac{f(\theta')}{f(\theta_j)} \times \frac{q(\theta_j|\theta')}{q(\theta'|\theta_j)}\right). \tag{3}$$
4. Set $\theta_{j+1} = \theta'$ with probability $\alpha$ and to $\theta_{j+1} = \theta_j$ otherwise.

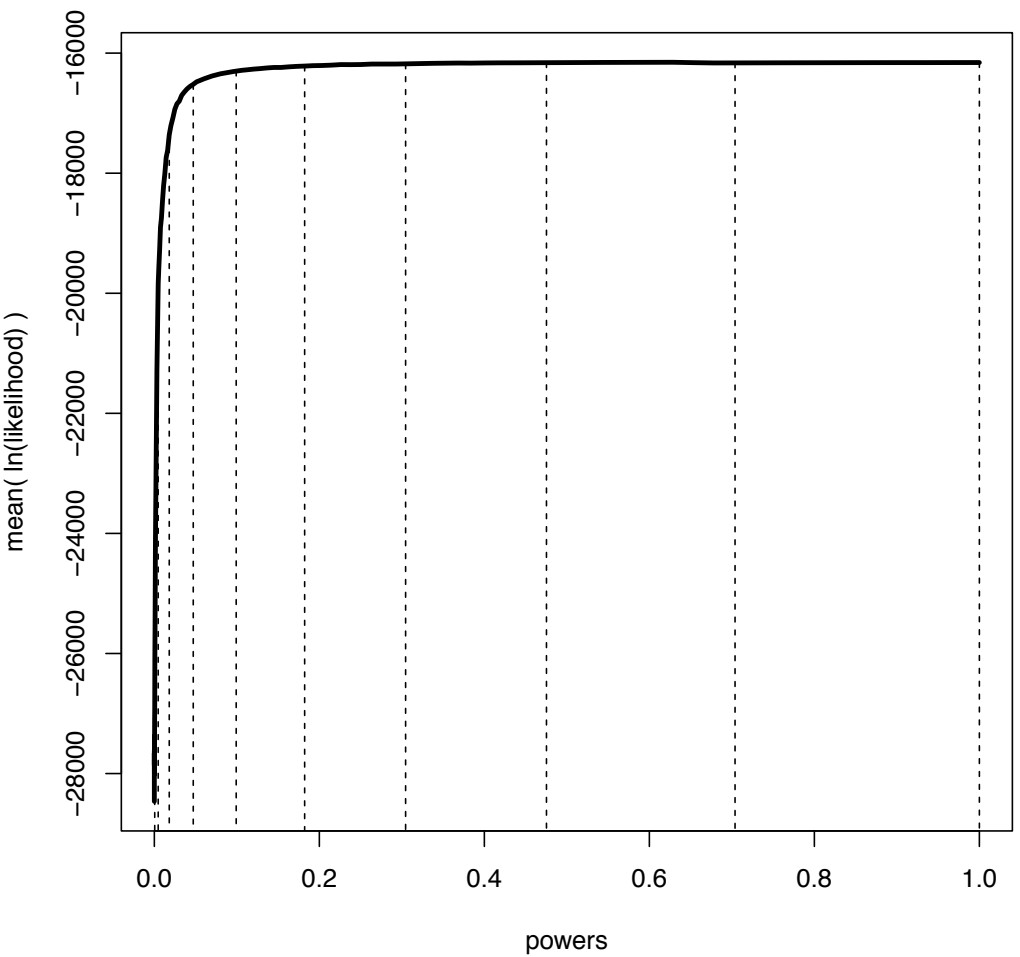

**Figure 1** **An example curve of mean log-likelihood samples over a range of different powers.** The vertical, dashed lines show which values of powers were used when $K = 11$ and $\beta_i = (i/(K-1))^{1.0/0.3}$ for $i \in \{0, K-1\}$. The curve shows explicitly over which range of powers the log-likelihood changes most drastically; when $\beta$ is small and thus the importance distribution is close to the prior. Hence, a good numerical approximation of the log-likelihood curve is obtained when most powers take small values.

As can be seen from this brief description of the modified MCMC algorithm, only the likelihood values need to be raised to the power $\beta_i$. All remaining aspects of the MCMC algorithm stay the same as the standard implementations in Bayesian phylogenetics (*Huelsenbeck & Ronquist, 2001*; *Drummond & Rambaut, 2007*; *Lakner et al., 2008*; *Lartillot, Lepage & Blanquart, 2009*; *Höhna & Drummond, 2012*).

It is important to note that every MCMC simulation for each power $\beta_j \in \beta$ necessarily includes its own burn-in period before the first sample can be taken. The power posterior analysis can be ordered to start from the full posterior ($\beta_{K-1} = 1.0$) and then to use monotonically decreasing powers until the prior ($\beta_0 = 0.0$) has been reached. Thus, the last sample of the previous power posterior run can be used as the new starting state. This strategy has been shown to be more efficient because it is easier to disperse from the
(concentrated) posterior distribution to the (vague) prior distribution thereby reducing the burn-in period significantly (*Baele et al., 2012a*).

## Parallel power posterior analyses

The sequential algorithm of a power posterior analysis starts with a pre-burnin phase to converge to the posterior distribution. Then, consecutive power posterior simulations are performed sequentially, starting with $\beta_{K-1} = 1.0$ (*i.e.,* the posterior) to $\beta_0 = 0.0$ (*i.e.,* the prior). Each power posterior simulation contains $L$ iterations, with the likelihood of the current state recorded every $T$th iteration. These 'thinned' samples are less correlated than the original draws from the MCMC simulation. The number of samples taken per power is $n = L/T$. At the beginning of each run a short burn-in phase is conducted, for example 10% or 25% of the run length.

The parallel algorithm for a power posterior analysis is set up almost identically to the sequential algorithm (see Fig. 2). Let us assume we have $M$ CPUs available. Then, we split the set of powers into $M$ consecutive blocks; the $m$th block containing the powers from $\beta_{\lfloor *K - \frac{(m-1)}{M}K - 1\rfloor}$ to $\beta_{\lfloor *K - (mK/M)\rfloor}$, *e.g.,* the first out of four blocks for 128 analyses contains $\{\beta_{127}, \ldots, \beta_{96}\}$, the second block contains $\{\beta_{95}, \ldots, \beta_{64}\}$, etc. If the set of $\beta$ cannot be split evenly into blocks then some blocks have one additional simulation, which is enforced by using only the integer part of the index. This block-strategy ensures that each CPU works on a set of consecutive powers which has the advantage of a shorter burn-in between simulations because the importance distributions are more similar to one another.

Regardless, each parallel sampler needs to start with an independent pre-burnin phase which creates an additional overhead. Thus, instead of running only one pre-burnin phase, as under the sequential power posterior analysis, we need to run $M$ pre-burnin phases. This overhead could be removed only if it would be possible to draw initial values directly from the power posterior distribution.

Figure 2 shows a schematic of our parallelization algorithm. After the initial pre-burnin phase, the workload is divided into blocks and equally distributed over the available CPUs. Note that CPUs can be combined for distributed likelihood computation. No synchronization or communication between samplers is necessary because each power posterior simulation is independent. The only parallelization barrier occurs at the end when all power posterior simulations have finished. Finally, the master CPU collects all likelihood samples, combines the results, and computes the marginal likelihood using one of equations given below. These equations are computationally cheap compared with obtaining the likelihood samples. We thus expect that the performance gain is close to linear with the number of available cores. The algorithm described here is implemented in the open-source software RevBayes (*Höhna et al., 2014*; *Höhna et al., 2016*), available at http://www.RevBayes.com.

Our implementation in RevBayes uses the *Message Passing Interface* (MPI). That is, a RevBayes instance that was compiled using MPI can be used on any standard Unix based computer or high performance cluster (HPC) and executed in parallel. From the user perspective, no additional commands between using the standard version of RevBayes and using the MPI version are needed. For example, the command powerPosterior(mymodel,
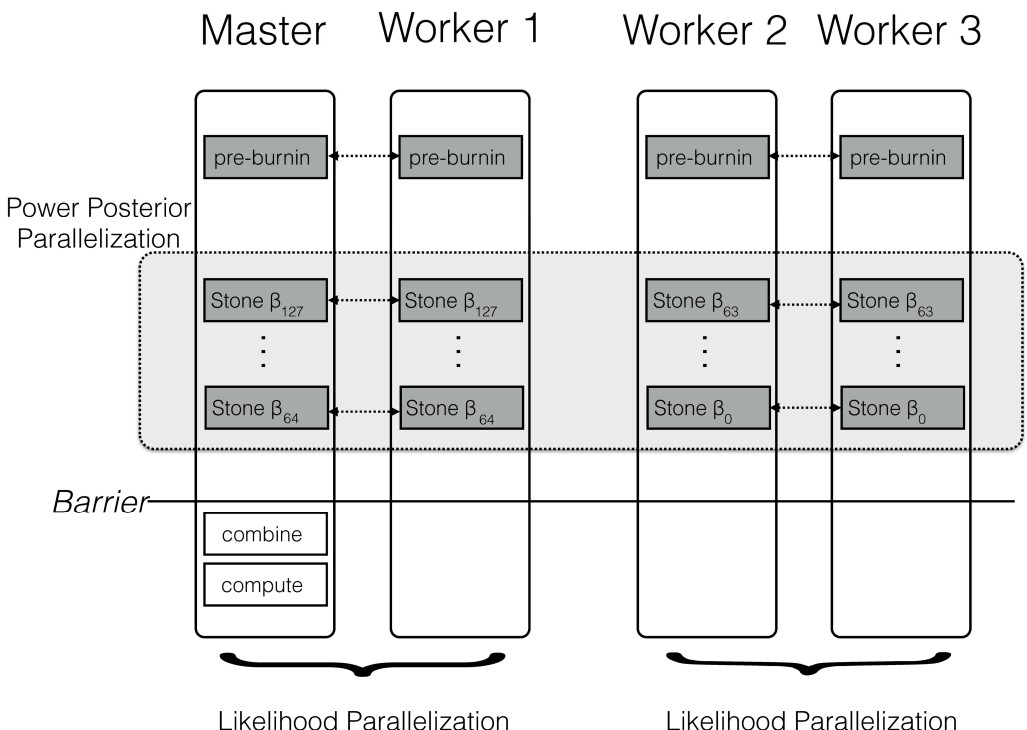

**Figure 2** **Schematic of the parallelization and workload balance between the master CPU and the worker CPUs.** In this example we have $M = 4$ CPUs and $K = 128$ power posterior simulations (stones). The first CPU is the designated master and the remaining CPUs are the workers/helpers. The power posterior simulations are divided into two blocks from $\beta_{127}$ to $\beta_{64}$ and $\beta_{63}$ to $\beta_0$. The first two CPUs work on the first block of power posterior simulations and the last two CPUs work on the second block. Each pair of CPUs shares the likelihood computation between them. Each CPU starts with its own pre-burnin phase. Then, each CPU runs its block of power posterior simulations. Finally, the master combines the likelihood samples and computes the marginal likelihood estimate. Thus, the only barrier is after all the single power posterior simulations, which is after each single CPU has finished its respective job.

moves, monitors, ``output/powers.out'', cats=100, sampleFreq=10) will automatically perform the power posterior analysis—as used in the study—in parallel. Hence, our implementation in RevBayes takes care of the parallelization for the user without the need of further specifications, assuming RevBayes was executed using MPI and several processes (*e.g.,* mpirun -np 16 rb myscript.Rev). Such an analyses can be run on any standard HPC with submission systems such as SLURM and TORQUE.

## Path-sampling

Path-sampling was the first numerical approximation method developed for marginal likelihood computation in Bayesian phylogenetic inference (*Lartillot & Philippe, 2006*). Path-sampling uses the trapezoidal rule to compute the integral of the log-likelihood samples between the prior and the posterior (see Fig. 1), which equals the marginal likelihood (*Lartillot & Philippe, 2006*). The equation of the trapezoidal rule for a single

likelihood sample from each power posterior simulation is

$$\ln f(D|M) = \sum_{k=0}^{K-1} \frac{(\ln(l_k) + \ln(l_{k+1})) * (\beta_{k+1} - \beta_k)}{2}. \tag{4}$$

Samples of the log-likelihood have a large variance. Hence, it is more robust to take many log-likelihood samples and use the mean instead. This yields the equation to estimate the marginal log-likelihood,

$$\ln f(D|M) = \sum_{k=0}^{K-1} \frac{\left(\frac{\sum_{i=1}^{n} \ln(l_{k,i})}{n} + \frac{\sum_{i=1}^{n} \ln(l_{k+1,i})}{n}\right) * (\beta_{k+1} - \beta_k)}{2} \tag{5}$$

which was proposed by *Baele et al. (2012a)*.

### Stepping-stone-sampling

Stepping-stone-sampling approximates the marginal likelihood by computing the ratio between the likelihood sampled from the posterior and the likelihood sampled from the prior. However, this ratio is unstable to compute and thus a series of intermediate ratios is computed: the stepping-stones (*Xie et al., 2011*; *Fan et al., 2011*). The stepping-stones can be chosen to be exactly the same powers as those used for path-sampling. The equation to approximate the marginal likelihood using stepping stone sampling is

$$f(D|M) = \prod_{k=0}^{K-1} \left( \frac{1}{n} \sum_{i=1}^{n} \frac{l_{k,i}^{\beta_{k+1}}}{l_{k,i}^{\beta_k}} \right)$$
$$= \prod_{k=0}^{K-1} \left( \frac{1}{n} \sum_{i=1}^{n} l_{k,i}^{\beta_{k+1} - \beta_k} \right) \tag{6}$$

Numerical stability of the computed marginal likelihood can be improved by retrieving first the highest log-likelihood sample, denoted by $\max_k$, for the $k$th power. Re-arranging Eq. (6) accordingly yields

$$\ln(f(D|M)) = \sum_{k=0}^{K-1} \left[ \ln \left( \sum_{i=1}^{n} \frac{\exp\left((\ln(l_{k,i}) - \max_k) * (\beta_{k+1} - \beta_k)\right)}{n} \right) + (\beta_{k+1} - \beta_k) * \max_k \right]. \tag{7}$$

As seen in Eqs. (5) and (7), only the set of likelihood, or log-likelihood, samples is needed to approximate the marginal likelihood. Both marginal likelihood estimates approach the true marginal likelihood when the number of samples and powers increases. Since both computations are comparably fast, they can be applied jointly and, for example, be used to test for accuracy without additional time requirements.

### Simulation design

Several previous studies have evaluated and compared the accuracy of different approaches to estimate the marginal likelihood of phylogenetic models (*Baele et al., 2012a*; *Baele, Lemey & Suchard, 2016*; *Oaks et al., 2019*). Overall, the findings suggest that power posterior based estimators (PS, SS and GSS) outperform other estimates for the cost of being
computationally more expensive. The objective of this simulation study was to test the performance gain when using multiple CPUs. Thus, we tested the performance of the parallel power posterior analyses using two phylogenetic examples; a smaller and a larger dataset. As the small example dataset we chose 23 primate species representing the majority of primate genera. We used only a single gene sequence, the cytochrome b subunit, containing 1141 base pairs. For the larger example data set we chose an alignment with 4 genes (with a total of 6720 base pairs) from 305 taxa of the superfamily *Muroidea* (*Schenk, Rowe & Steppan, 2013*). For both examples we used the same model with the only difference that the larger dataset was partitioned into four subsets of sites (see protocols 1 and 2 from *Höhna, Landis & Heath, 2017*). We assumed that molecular evolution can be modeled by a general time reversible (GTR) substitution process (*Tavaré, 1986*) with four gamma-distributed rate categories (*Yang, 1994*). Furthermore, we assumed a strict, global clock (*Zuckerkandl & Pauling, 1962*) and calibrated the age of the root. As a prior distribution on the tree we used a constant-rate birth-death process with diversified taxon sampling (*Höhna et al., 2011*; *Höhna, 2014*) motivated by the fact that one representative species per genus was sampled, which is clearly a non-random sampling approach. The specific models correspond to the protocols described in *Höhna, Landis & Heath (2017)* and can also be found as tutorials at https://revbayes.github.io/tutorials/.

Each analysis consisted of a set of $K = 100$ power posterior simulations (see Fig. 2 for a schematic overview). The analyses started with a pre-burnin period of 10,000 iterations to converge to the posterior distribution. Note that in RevBayes each MCMC iterations consists of several moves (compared with MrBayes (*Ronquist et al., 2012*) and BEAST (*Suchard et al., 2018*) where each iteration consists of only a single move); the primates analysis included 38 moves and the *Murdoidea* analysis included 73 moves per iteration. Then, each power posterior analysis was run for 10,000 iterations and samples of the likelihood were taken every 10 iterations. The 25% initial samples of each power posterior distribution were discarded as additional burnin. The marginal likelihood was estimated using both path-sampling and stepping-stone-sampling once all power posterior simulations had finished as they contribute to performance overhead in practice. We ran each analysis 10 times and measured the computation time on the San Diego Supercomputer (SDSC) Gordon. Each compute node on Gordon contains two 8-core 2.6 GHz Intel EM64T Xeon E5 (Sandy Bridge) processors. The experiment was executed using 1, 2, 4, 8, 16, 32 and 64 CPUs, respectively. For each number $k$ of CPUs used, we repeated the analyses by assigning 1, 2, 4,...64 CPUs to parallelizing the likelihood computation instead of distributing the stones. Thus, we additionally tested if parallelization over stones, the likelihood computation, or a mixture is most efficient.

## RESULTS

We present the results of the average runtime as a function of the number of CPUs used in Fig. 3. Performance gains are most pronounced when few CPUs are used. The runtime is almost halved when compared between 1 and 2 CPUs or 2 and 4 CPUs. For example, our primate analyses took on average 11.68 h when using only a single CPU. By contrast, the

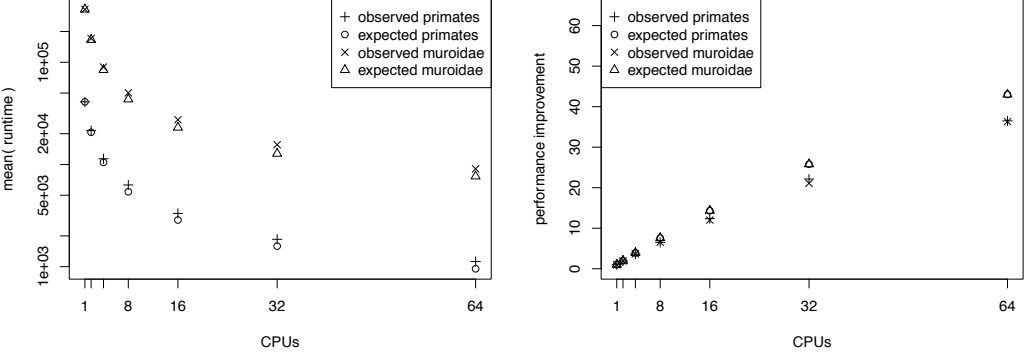

**Figure 3** **The average runtime of a marginal likelihood estimation on a simple phylogenetic model recorded over 10 repeated runs.** The analyses were performed on the San Diego supercomputer cluster Gordon using 1, 2, 4, 8, 16, 32 and 64 CPUs. The runtimes were measured in seconds. The left graph shows the mean runtime as a function of the number of CPUs. The right graph shows the performance increase (fraction of time needed) compared with a single CPU. Both graphs show the actual performance increase and the expected performance increase (if there were no overhead between CPUs).

**Table 1** **Runtime using $M$ CPUs (rows) of which $N$ CPUs (columns) are assigned to the likelihood computation. Here we show the results of the primates dataset.**

| $M \setminus N$ | 1 | 2 | 4 | 8 | 16 | 32 | 64 |
|---|---|---|---|---|---|---|---|
| 1 | 42,063 | * | – | – | – | – | – |
| 2 | 21,768 | 22,268 | – | – | – | – | – |
| 4 | 11,275 | 11,434 | 12,088 | – | – | – | – |
| 8 | 6,253 | 6,136 | 6,185 | 6,969 | – | – | – |
| 16 | 3,336 | 3,189 | 3,162 | 3,562 | 4,612 | – | – |
| 32 | 1,856 | 1,709 | 1,651 | 1,846 | 2,393 | 4,738 | – |
| 64 | 1,112 | 944 | 880 | 966 | 1,217 | 2,406 | 11,363 |

analyses took only 6.04 h and 3.13 h when we used two CPUs and four CPUs respectively (Table 1). Virtually the same runtime improvements were achieved for the larger *Murdoidea* dataset (Fig. 3, Table 2).

The performance increase levels off quickly once 8 or 16 CPUs are used. This is simply due to the fact that twice as many CPUs are needed each time to roughly halve the computational time. Hence, the gain from 1 to 4 CPUs is approximately equivalent to the gain from 16 to 64 CPUs. Furthermore, in our setup on Gordon each compute node had 16 CPUs which means that an additional communication overhead occurs once more than 16 CPUs are used. Additionally, the overhead (*i.e.,* the independently run pre-burnin for each chain) which each CPU needs to perform reduces the performance gain for larger number of CPUs.

We computed the expected runtime to assess whether our implementation achieved the largest possible performance gain. For example, we wanted to explore if there is an additional overhead for using parallelization that was possibly introduced by our specific implementation. Having $M$ CPUs available, each CPU needs to run at most $\lceil K/M \rceil$ power

**Table 2  Runtime using *M* CPUs (rows) of which *N* CPUs (columns) are assigned to the likelihood computation. Here we show the results of the *Muroidea* dataset.**

| M \ N | 1 | 2 | 4 | 8 | 16 | 32 | 64 |
|---|---|---|---|---|---|---|---|
| 1 | 329,246 | – | – | – | – | – | – |
| 2 | 171,797 | * | – | – | – | – | – |
| 4 | 92,858 | 92,212 | 90,432 | – | – | – | – |
| 8 | 52,329 | 51,072 | 50,244 | 53,411 | – | – | – |
| 16 | 28,248 | 26,426 | 26,792 | 27,573 | 29,297 | – | – |
| 32 | 15,418 | 14,423 | 14,126 | 14,599 | 15,371 | 18,450 | – |
| 64 | 9,365 | 8,234 | 7,705 | 7,649 | 8173 | 9,641 | 18,272 |

**Notes.**

*Runs using $M = 2$ CPUs with $N = 2$ CPUs per likelihood did not finish within the wall-time provided by XSEDE.

posterior simulations, which is the ratio of the total number of power posterior simulations to CPUs rounded upwards (ceiling). Additionally, each CPU runs its own pre-burnin phase, which had the same length as a single power posterior simulation in our tests. Therefore, we can compute the average runtime of a single power posterior simulation by dividing the runtime of the single CPU analysis by $K + 1$. Then, the expected runtime for $M$ CPUs, $t_M$, is given by

$$\mathbb{E}[t_M] = t_1 \times \frac{\lceil K/M \rceil + 1}{K + 1} \tag{8}$$

where $t_1$ corresponds to the runtime when only one CPU was available. In general, our implementation seems to perform close to the expected optimal performance (Fig. 3). However, we observe an increasing discrepancy between the expected and the observed performance gain when many CPUs were used. This discrepancy is most likely due to bottlenecks in competing hardware allocations. For example, we noticed that I/O operations performed on a network filesystem, which are commonly used among large computer clusters, significantly influenced the performance, especially when many CPUs frequently wrote samples of the parameters to a file. To further improve our implementation, this minor problem could be alleviated by standard approaches such as async/non-blocking I/O.

We performed an additional performance analysis where we omitted the pre-burnin phase (running the MCMC sampler on the posterior distribution, see Fig. 2) but kept the burnin phase for each power posterior distribution. This scenario could be realistic when one has already performed a full posterior probability estimation and only wants to compute the marginal likelihoods for model selection. In this case, the samples from the posterior distribution can be used to specify starting values of the power posterior analysis. Here we see that the performance improvement becomes more linear with the number of CPUs (see Fig. 4). Although this case might not happen frequently in practice, we use this to demonstrate that only the pre-burnin phase prevents us from having an almost linear, and thus optimal, performance increase.

We also investigated whether the performance overhead (observed in Fig. 3) is correlated with the number of stepping stones per CPU. For example, we observed the largest difference

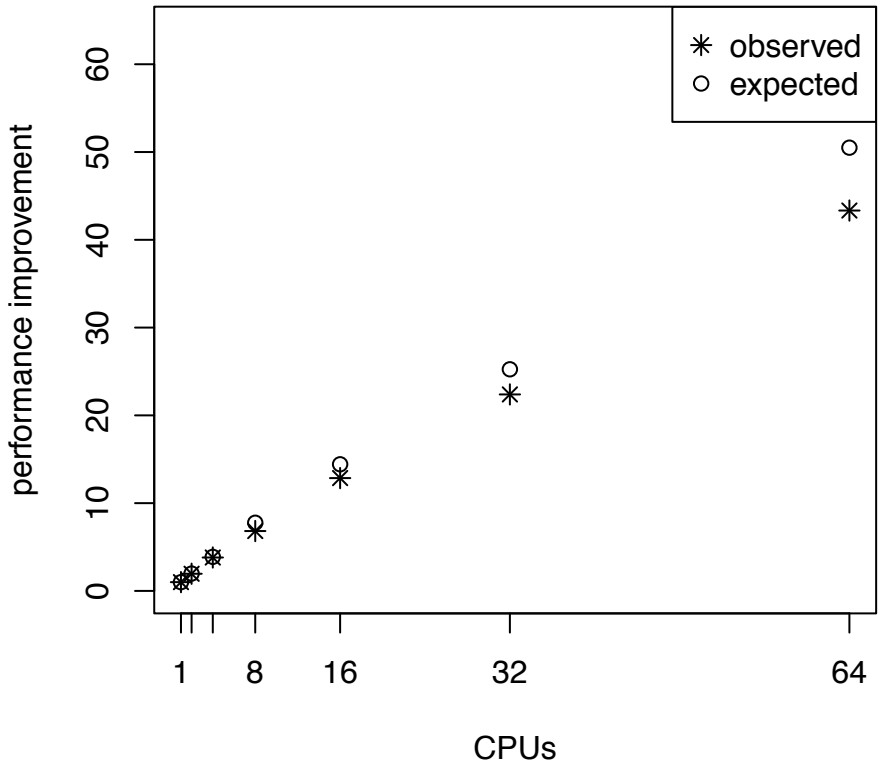

**Figure 4** **The average performance improvement (runtime reduction) when estimating the marginal likelihood on a simple phylogenetic model *without* pre-burnin phase, recorded over 10 repeated runs.** The analyses were performed on the San Diego supercomputer cluster Gordon using 1, 2, 4, 8, 16, 32 and 64 CPUs. The runtimes were measured in seconds. The graph shows the actual and the expected performance increase compared with a single CPU, where performance is nearly linear.

between the expected and actual runtime when 64 CPUs were used (each CPU ran only one or two power posterior simulations plus the pre-burnin phase). Thus, we tested if there was an effect of small numbers of power posterior simulations by running analysis with $K \in \{2, 3, 5, 10, 20, 30, 40, 50\}$ on a single CPU. As the expected runtime, we computed the mean runtime per individual power posterior simulation when $K = 50$. Our results, shown in Fig. 5, demonstrate that there is an intrinsic overhead for small number of power posterior simulations. This overhead seemed to be the cause of the discrepancy between our expected and observed performance increase in the parallel power posterior algorithm (Fig. 3). Part of the overhead is caused by the additional time to start the process, load the data, allocate memory, receive file handles and all other tasks that need to be performed before and after a power posterior analysis.

Finally, we compared the performance increase when parallelizing the power posterior analysis, the likelihood computation, or both. For this combined parallelization scheme we implemented a hierarchical parallelization structure as describe by *Aberer, Kobert & Stamatakis (2014)*. For example, when 4 CPUs are available we can divide the likelihood computation over 2 CPUs and divide the power poster analysis into 2 blocks (see Fig. 1). This

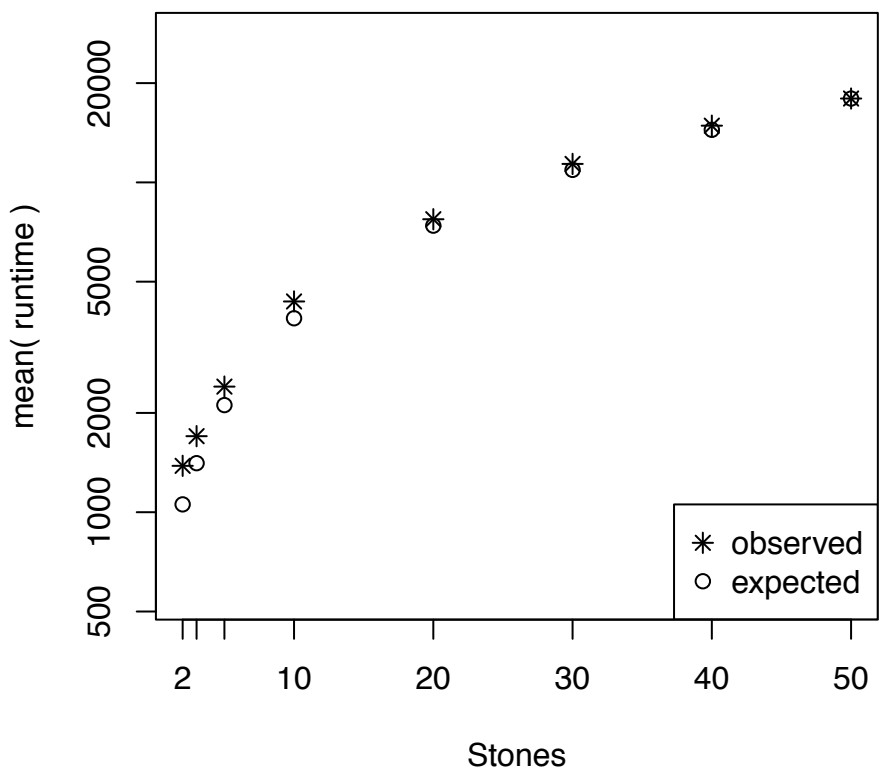

**Figure 5** **The average runtime over 10 repeated runs of a marginal likelihood estimation on a simple phylogenetic model for different number of powers posterior simulations $K$.** The runtimes were measured in seconds. The graph shows the actual runtime and the expected runtime which is based on the mean runtime per power posterior simulation when $K = 50$.

test thus includes the parallelization approach over the likelihood function as suggested by *Baele & Lemey (2013)*. We tested the performance difference using $M = \{2, 4, 8, 16, 32, 64\}$ CPUs of which we assigned $N$ to share the likelihood computation. We observed the best overall runtime reduction when we applied a combined likelihood and power posterior analysis parallelization (Tables 1 and 2). Furthermore, the improvement of each parallelization yields diminishing returns when many CPUs are used, which additionally supports the utility of a combined parallelization scheme. We conclude that using $N = \lfloor \sqrt{M} \rfloor$ will give the overall best performance and set this distribution of CPUs as the default option in RevBayes. Nevertheless, the best performance will depend on the specific dataset (*i.e.,* number of base pairs), the number of power posteriors ( $K$) and the pre-burnin length. A motivated user can manually set how many CPUs are used to parallelize the likelihood computation (option procPerLikelihood in the powerPosterior command) although the specific speedup might be minor compared with the overall improvement (Tables 1 and 2).

## CONCLUSION

Modern phylogenetic analyses depend on increasingly complex models and increasingly large data set sizes. Even phylogenetic analyses which do not use molecular sequence data (for example, diversification rate analyses (*FitzJohn, 2012*), continuous trait analyses (*Uyeda & Harmon, 2014*), and historical biogeography analyses (*Landis et al., 2013*) have grown more complex and use time-intensive likelihood calculations that are not always easily parallelizable. Both trends lead to longer runtimes, which is even more pronounced for Bayesian model selection exercises using marginal likelihoods (*Oaks et al., 2019*; *Fourment et al., 2020*); the path-sampling and stepping-stone-sampling algorithms used for approximating marginal likelihoods are inherently computationally demanding. In the present paper we have developed a simple parallel algorithm to speed up the computation of marginal likelihoods for Bayesian phylogenetic inference. In our simulation study, which serves mostly as a proof of concept, we showed that performance improvement is close to linear for few CPUs, *i.e.,* between one and 16 CPUs. An analysis that previously took 8 weeks on a single CPU can now be completed in four days when 16 CPUs are available.

   Current implementations of power posterior algorithms are either sequential or parallelize the likelihood computation (*Ayres et al., 2019* *e.g.,* using the BEAGLE library). Our new parallel power posterior analysis can be more than an order of magnitude faster than ordinary, sequential algorithms. The presented parallel algorithm should be straightforward to be implemented in other software or applied to a variety of different model types. For example, marginal likelihood estimation using the software package BEAST can be manually modified to perform each step of the power posterior simulation independently *Baele, Van de Peer & Vansteelandt (2009)*; *Suchard et al. (2018)*, and thus each power posterior could be computed in parallel, but is not automated for the user. Finally, the described parallelization scheme should be applicable to alternative methods for computing marginal likelihood (*e.g.,* the generalized stepping-stone sampler (GSS) *Fan et al., 2011*; *Holder et al., 2014*; *Baele, Lemey & Suchard, 2016*) and Bayes factors directly (*Lartillot & Philippe, 2006*; *Baele, Lemey & Vansteelandt, 2013*) because all these approaches rely on a set of power posterior analyses.

## ACKNOWLEDGEMENTS

Alfonso Valencia, Alexandros Stamatakis, Guy Baele, Alexey Kozlov and an anonymous reviewer provided comments that improved an earlier version of this manuscript.

### Funding

This work was supported by Deutsche Forschungsgemeinschaft (DFG) Emmy Noether-Program HO 6201/1-1. The funders had no role in study design, data collection and analysis, decision to publish, or preparation of the manuscript.

## Grant Disclosures
The following grant information was disclosed by the authors:
Deutsche Forschungsgemeinschaft (DFG) Emmy Noether-Program:  HO 6201/1-1.

## Competing Interests
The authors declare there are no competing interests.

## Author Contributions
- Sebastian Höhna conceived and designed the experiments, performed the experiments, analyzed the data, prepared figures and/or tables, authored or reviewed drafts of the paper, and approved the final draft.
- Michael J. Landis conceived and designed the experiments, analyzed the data, authored or reviewed drafts of the paper, and approved the final draft.
- John P. Huelsenbeck conceived and designed the experiments, authored or reviewed drafts of the paper, and approved the final draft.

## Data Availability
The source code of the RevBayes is available at Github and includes an archive with the specific scripts and data for the experiments (see readme for details: https://github.com/revbayes/revbayes.

## Supplemental Information
Supplemental information for this article can be found online at http://dx.doi.org/10.7717/peerj.12438#supplemental-information.

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
