# Peer review of "Parallel power posterior analyses for fast computation of marginal likelihoods in phylogenetics"

_PeerJ, doi:10.7717/peerj.12438_

## Round 0.1 · original submission · Minor Revisions

I have now received two reviews of your paper. I am happy to report that both reviewers found your paper interesting and appropriate for the journal. Both, however, have a number of suggestions to help improve your paper. Therefore, I am recommending 'minor revision' so that you can address these suggested edits and clarifications. Congratulations.

·

Basic reporting

The authors propose a parallellisation of the PS and SS approaches to decrease computation time. For these approaches, the marginal likelihood readily splits up into independent bits that can be estimated and combined into the overall (log) marginal likelihood. The implementation and the reported performance increases are certainly interesting for these demanding estimations, but the authors need to put a bit more work into correctly mentioning other publications regarding similar work and a better estimator such as GSS.

Otherwise, the manuscript is well written and easy to read through. The figures are clear but do not offer any insights into the accuracy of the reported estimates.

Specific questions:

Lines 14-15: A more general statement would be preferred, as both these methods have slightly less impressive properties than generalized stepping-stone sampling (GSS). It’s a bit strange that GSS isn’t mentioned by name in the introduction as I think it should be, as it related to how generalizable the approach of the authors actually is. The working priors required for GSS can only be constructed from an actual posterior exploration run, and not from a power posterior pre-burnin analysis. The authors need to be clear on this issue and mention it in the manuscript. Suggested citations: Fan et al. (2011); Holder et al., 2014; Baele et al., 2016.

Experimental design

The experimental design seems correct for the task at hand, i.e. measuring performance increases through the parallel implementation. No design was set up to measure correctness again a known value in my opinion. As an alternative, the authors could compare the results against those of a single-instance analysis, although that would a comparison against an approximation / estimate.

Specific questions:

Line 168: what about a simulation study for correctness? The GSS publications have already shown that PS and SS generate biased estimates of the true (log) marginal likelihood. Performance is certainly important, but I would expect that accuracy is still the more important aspect. It would be interesting to see how much accuracy is lost (if any) when opting for maximum performance.

Line 185-189: these are rather low settings and I don’t think these should be applied to any empirical data set analyses; sampling every 10th iteration is too often in my opinion and a chain length of at least 100,000 but preferable longer will be required (I showed this in a 2014 book chapter I believe: “Bayesian model selection in phylogenetics and genealogy-based population genetics”).

Lines 223-229: This requires some more explanation, especially lines 225-226. Values from a posterior analysis are not guaranteed to offer good starting values for each power posterior analysis as the closer you get to the prior, the more these values can move away from their posterior estimates.

Validity of the findings

The findings are well represented but the overall manuscript is a bit lacking in terms of citations and acknowledging similar efforts.

This type of work has been done before, although it will be good to have this available in RevBayes. You can manually edit BEAST XML files to have BEAST estimate these separate parts of the (log) marginal likelihood as well, although it’s not fully automated to launch the separate jobs and combine the end result. Still, I think it may be good to mention that BEAST 1.10 has this ability to compare the MLE contribution of a range of powers, or just one single power at a time (although there's probably no documentation about how to do it).

The main novelty of this manuscript is to be found in the actual implementation in RevBayes. I think I may have been the first to split these types of marginal likelihoodestimations into multiple pieces although I can’t fault the authors for not knowing about this particular publication (https://bmcecolevol.biomedcentral.com/articles/10.1186/1471-2148-9-87). The authors do need to make it clear how automated such a parallel analysis is from the end user’s point of view. Is a single command required to run such a parallel analysis? How portable is the implementation across different server systems, i.e. regular servers and/or job/queue-based systems?

Additional comments

I appreciate the work being done here, and certainly the estimations are cumbersome and can benefit from faster implementations. It would have been nice to show that this approach enables tackling problems that were previous not possible though, although this is quite intuitive as well. I am in favor of acceptance provided that the authors put some more efforts into acknowledging past efforts.

·

Basic reporting

1. Is power posterior implemented by tools other than RevBayes? If so, are those implementations parallelized? How do they compare in terms of performance?
2. Raw data not shared (datasets, scripts/command lines, outputs)

Experimental design

1. How was parallelization implemented technically? MPI, OpenMP, hybrid?
2. How many sites does Muroidea dataset have? This value is crucial for the scalability of site-based likelihood parallelization (as in Aberer 2014).
3. Please provide hardware details for Gordon cluster (CPU model, number of cores). If multiple cluster nodes were used in some configurations (e.g. for 32/64 CPUs), it probably had impact on respective performance improvements (e.g. higher communication overhead, but higher memory bandwidth etc.). So please clarify/discuss this.
4. In Fig.6 (speedups w/o pre-burnin), how expected speedup values were computed? E.g., with 64 CPUs the expected speedup is only ~50x, why?

Validity of the findings

1. Number of iterations in pre-burnin and each power analysis are both set to 10,000. Is this common setup in practice? Since pre-burnin represents the sequential bottleneck, changing the ratio between number of iterations in both phases will substantially change scalability of parallelization.
2. Since combined parallelization yields best results (and used by default in RevBayes), it would make sense to focus on combined speedups in reporting (abstract, plots). Currently, it is not clear whether 11.4x speedup on 16 CPUs in abstract refers to combined parallelization or not?

Additional comments

1. L219: If I/O operations do indeed represent a bottleneck for parallelization, this problem can be by alleviated by standard approaches such as async/non-blocking I/O (=thread does not have to wait until write operation is completed)
2. With combined parallelization scheme, the optimal layout will probably depend on the i) number of sites (bottleneck for likelihood parallelization) and ii) number of steps (K) and iterations (bottleneck for power parallization). So using this information will likely yield a better prediction than N=sqrt(M).

Minor:
- tables 1 and 2 belong together, maybe just merge this info into one table. Also, please indicate best runtime per row with bold/color font.
- please consider using color to make plots more readable (PeerJ is an online journal, so no need to optimize for print)

---

## Round 0.2 · accepted · Accept

I have sent your paper back to the two previous reviewers. Both have now reviewed your revised paper and we are all in agreement that your paper is ready for publication. We are recommending acceptance. Congratulations and thank you for submitting your exciting work to PeerJ.

·

Basic reporting

This revised version of the manuscript is much improved in terms of acknowledging past efforts, and overall looks very nice. I thank and commend the authors for responding in detail to my questions and comments.

Experimental design

No further comments.

Validity of the findings

The authors focus on the performance improvements and provide references pertaining to the accuracy of these methods. As the authors state, they do not offer a novel method but instead a much needed improvement in performance of an existing marginal likelihood estimation method.

Additional comments

No further comments.

·

Basic reporting

Thanks for addressing my comments.

Experimental design

Thanks for addressing my comments.

Validity of the findings

Thanks for addressing my comments.